# Susceptibility of Four Abalone Species, *Haliotis gigantea*, *Haliotis discus discus*, *Haliotis discus hannai* and *Haliotis diversicolor*, to Abalone asfa-like Virus

**DOI:** 10.3390/v13112315

**Published:** 2021-11-20

**Authors:** Tomomasa Matsuyama, Ikunari Kiryu, Mari Inada, Tomokazu Takano, Yuta Matsuura, Takashi Kamaishi

**Affiliations:** 1Research Center for Fish Diseases, National Research Institute of Aquaculture, Japan Fisheries Research and Education Agency, Minami-Ise 516-0193, Japan; takanoto@affrc.go.jp (T.T.); ymatsuura@affrc.go.jp (Y.M.); kamaishi@affrc.go.jp (T.K.); 2Diagnosis and Training Center for Fish Diseases, National Research Institute of Aquaculture, Japan Fisheries Research and Education Agency, Minami-Ise 516-0193, Japan; ikunari@affrc.go.jp (I.K.); inadamari@affrc.go.jp (M.I.)

**Keywords:** abalone asfa-like virus, AbALV, abalone, Haliotis, virulence, infectivity, mass mortality

## Abstract

Abalone amyotrophia is a viral disease that causes mass mortality of juvenile *Haliotis discus* and *H. madaka*. Although the cause of this disease has yet to be identified, we had previously postulated a novel virus with partial genome sequence similarity to that of African swine fever virus is the causative agent and proposed abalone asfa-like virus (AbALV) as a provisional name. In this study, three species of juvenile abalone (*H. gigantea*, *H. discus discus*, and *H. diversicolor*) and four species of adult abalone (the above three species plus *H. discus hannai*) were experimentally infected, and their susceptibility to AbALV was investigated by recording mortality, quantitatively determining viral load by PCR, and conducting immunohistological studies. In the infection test using 7-month-old animals, *H. gigantea*, which was previously reported to be insusceptible to the disease, showed multiplication of the virus to the same extent as in *H. discus discus*, resulting in mass mortality. *H. discus discus* at 7 months old showed abnormal cell masses, notches in the edge of the shell and brown pigmentation inside of the shell, which are histopathological and external features of this disease, while *H. gigantea* did not show any of these characteristics despite suffering high mortality. Adult abalones had low mortality and viral replication in all species; however, all three species, except *H. diversicolor*, became carriers of the virus. In immunohistological observations, cells positive for viral antigens were detected predominantly in the gills of juvenile *H. discus discus* and *H. gigantea*, and mass mortality was observed in these species. In *H. diversicolor*, neither juvenile nor adult mortality from infection occurred, and the AbALV genome was not increased by experimental infection through cohabitation or injection. Our results suggest that *H. gigantea, H. discus discus and H. discus hannai* are susceptible to AbALV, while *H. diversicolor* is not. These results confirmed that AbALV is the etiological agent of abalone amyotrophia.

## 1. Introduction

Abalones are an important commercial fisheries resource with high market value. However, drastic declines in fishery production of abalone have been documented worldwide [1]. In Japan, releasing hatchery-grown seed stock into natural habitats has been conducted nationwide since the late 1960s in order to restore diminished abalone stocks [2]. Nevertheless, abalone continues to decline in most regions of Japan [2].

As with many other farmed organisms, a variety of infectious diseases have developed in the high-density production of abalone seedlings. In Japan, these include *Vibrio harveyi* [3,4], *Francisella halioticida* [5,6], *Candidatus* Xenohaliotis californiensis [7], and a viral disease called abalone amyotrophia [8].

Abalone amyotrophia is an infectious disease named to reflect the atrophy of foot muscles in diseased abalones, a symptom that is associated with mass mortality [8] and has been observed since the early 1980s [9]. Diseased abalones have reduced attachment activity and sometimes form notches in the edge of the shell and brown pigmentation on the inside of the shell [10]. On histopathological observation, abnormal cell masses are mainly formed in the nerve trunk and peripheral nerves of muscle [10]. While abalone herpesviruses (AbHV) [11] and Shriveling Syndrome Associated Virus (AbSV) [12] have been reported to cause acute disease with rapid onset and high mortality regardless of age, abalone amyotrophia causes chronic mortality mainly in juveniles [13]. The causative agent was presumed to be a virus because homogenates obtained from affected abalone and filtered through a 0.22-μm filter were infectious [14,15]. Several efforts have been made to discover the pathogen by electron microscopy [10,16] and virus isolation [17], however no positive identification of the causative agent was found for a long time.

In a previous report, we semi-purified the pathogen of abalone amyotrophia and analyzed the genome of the purified fraction using next-generation sequencing (NGS). Epidemiological surveys were conducted on diseased and healthy abalones by RT-PCR and conventional PCR targeting contigs obtained by assembling short reads of NGS. A sequence of about 155 kbp, which was considered to be a part of the pathogen genome because of its high specificity for disease in abalones, contained several open reading frames with high homology to African swine fever virus (ASFV), suggesting that a virus closely related to ASFV is the pathogen. Therefore, we provisionally proposed designating the putative causative agent of abalone amyotrophia as a new virus called abalone asfa-like virus (AbALV) [18]. The development of abalone amyotrophia has been confirmed experimentally in *H. discus discus* [9] and empirically in *H. discus hannai* and *H. madaka* [10]. In a previous report, the genome of AbALV was detected in *H. discus discus* and *H. madaka* [18], while its infectivity to other abalone species is unknown. It is also known that abalone amyotrophia is a disease of juvenile and causes less mortality in grown abalones [13]. However, since infection of abalone amyotrophia in juveniles was reported to be established by co-habitation with field collected spawners [19], it is suspected that adult abalone is a likely carrier of the virus. For these reasons, it is important to clarify the abalone species and age at which AbALV can infect them. In this study, we experimentally infected *H. discus discus*, *H. discus hannai*, *H. gigantea*, and *H. diversicolor* of different ages using effluent from containers of affected abalones as the source of infection and compared the infectivity and virulence of AbALV among abalone species and ages. In addition to monitoring mortality after experimental infection, changes in virus load and infected cells in abalone were measured by quantitative PCR (qPCR) and immunostaining with antisera prepared against recombinant major capsid protein (MCP, homolog of p72 gene of ASFV) of AbALV, respectively.

## 2. Materials and Methods

### 2.1. Ethics Statement

All experiments using mice and abalone in this study were carried out following the ARRIVE (Animal Research: Reporting of In Vivo Experiments) guidelines. Mouse handling, husbandry, and sampling were conducted based on the policy of the Institutional Animal Care and Use Committee (29 April 2018 and 27 March 2020) of the Fisheries Research and Education Agency under the approval of the committee (IACUC-NRIA nos. 20001, 20005 and 30003).

### 2.2. Abalones

The abalones shown in Table 1 were obtained from seed production facilities. Abalones aged 4 to 5 years old were obtained at 1 year of age and reared at the facility of the Japan Fisheries Research and Education Agency until use in experiments. No external abnormalities were observed in lots of healthy abalone. Before experiments, ten randomly sampled healthy abalones from each lot were screened using AbALV MCP gene-specific PCR [18], and all were negative. Survivors from incubation lots that showed mortality and were positive for the AbALV MCP gene by PCR were used to infect other abalones in infection experiment-1 (*H. gigantea*) and as a source of materials for immunohistological examination (*H. discus discus*). Abalones were fed with dried kelp (*Saccharina japonica*) once per week throughout the experiments. All tanks received aerated, UV-irradiated, flow-through seawater (approximately 300 mL/min). The water temperature during the experiment was 18–20 °C unless otherwise noted.

### 2.3. Infection Experiment 1: Cohabitation Infection with Four Species at Different Ages

The *H. gigantea* used as the source of infection were obtained in May 2019, and individuals of this lot had been dying since January of the same year, reaching a cumulative mortality rate of about 10% by the time of use in this experiment. DNA extracted from the shell muscles of 10 randomly selected abalones in this group tested positive for the MCP gene of AbALV by conventional PCR [18]. Twenty *H. gigantea* of this group were randomly collected and kept in a 27-L aquarium, and the effluent from the aquarium was distributed to the aquarium of the infected group for 3 days (shown in Figure 1). The amount of virus in the shell muscle of survived infectious source was measured by qPCR after 3 days.

Using the experimental set-up shown in Figure 1, the effluent from a tank containing infected *H. gigantea* abalone was distributed to 56-L tanks with 7-month-old *H. gigantea, H. discus discus*, and *H. diversicolor* with each tank containing 500, 100 and 100 individuals for each species and one 180-L tank containing 4- to 5-year-old *H. gigantea, H. discus discus, H. discus hannai,* and *H. diversicolor* (20 of each species). As a control (non-infected group), three 56-L tanks with 100 individuals of 7-month-old *H. gigantea, H. discus discus*, and *H. diversicolor* and one 180-L tank containing 4- to 5-year-old *H. gigantea, H. discus discus, H. discus hannai,* and *H. diversicolor* (20 of each species) were supplied with UV-irradiated seawater. Infected group tanks with 500 abalone were sampled, while the infected group and non-infected group tanks with100 abalones were observed to record mortality. The larger tanks with mixed species of 4- to 5-year-old abalone (infected and non-infected groups) were observed for mortality.

For the 7-month-old abalones, 20 individuals were collected from the sampling tank on 1, 3, 7, 14, 21, 28, 35, 42, 49, 56, and 63 days post-infection (dpi) to examine the kinetics of AbALV load, and 5 individuals were collected for immunohistological observations (tissue was not collected on day 63 because there were not enough abalone in the sampling group). For the 4- to 5-year-old abalones, 10 animals of each species were randomly collected from observation tanks at 63 dpi. Five individuals of each species were subjected to measurement of the viral load and for immunohistological observation. The amount of virus in tissues was determined by measuring the AbALV MCP gene copy number included in DNA extracted from the whole body of 7-month-old abalones and from shell muscle tissue and visceral mass (including gills, hypobranchial *glands*, intestines, kidneys, heart and surrounding connective tissue) of 4- to 5-year-old abalones. For immunohistological analysis, the whole body, including the shell of 7-month-old abalones and the muscle tissue, visceral mass, and head of 4- to 5-year-old abalone were sampled and fixed in Davidson’s fixative [20] and then stored in 70% ethanol.

### 2.4. Infection Experiment 2: Injection Infection of H. discus discus and H. diversicolor at 12 Months of Age

Diseased *H. discus discus* which was collected during a previous study [18] and stored at −80 °C was used as the source of infection. Approximately 0.4 g of soft-body tissue, excluding the midgut, was collected from three cryopreserved *H. discus discus*. The tissues were minced with a razor, combined with ten volumes of autoclaved seawater, and ground with a Potter-type glass homogenizer. The crushed samples were centrifuged at 15,000× *g* for 10 min at 4 °C, and the supernatant was passed through a 0.22-μm syringe filter (Merck Millipore, Germany). The filtrate was injected into the foot muscles of 50 *H. discus discus* or 140 *H. diversicolor* healthy abalones with a 30-gauge needle (20 μL/animal). Separately, 20 abalones of each species were reared in observation tanks. To determine the temporal changes in virus load, 5 *H. discus discus* and 20 *H. diversicolor* were randomly collected periodically from the sampling tanks. AbALV MCP gene copy number per microgram of total DNA in shell muscle tissues was determined.

### 2.5. Generation of Mouse Antisera against AbALV MCP and Piscine orthoreovirus 2 (PRV-2) σ-1 Recombinant Proteins

Recombinant N-terminal histidine-tagged AbALV MCP protein and piscine orthoreovirus 2 (PRV-2) [21] σ-1 protein was produced using *Escherichia coli*. To verify the specificity of the antiserum against AbALV MCP protein, the antiserum against PRV-2 σ-1 protein was used as a negative control. Specific primers with the In-Fusion recombination cloning sequence in the tail (Appendix A) were used to amplify the MCP gene and σ-1 gene from the start codon to the termination codon by PCR. The template for the MCP gene was created using DNA extracted from diseased *H. discus discus* in a previous study [18], and cDNA reverse transcribed in a previous study [21] was used as the template for the σ1 gene. PCR was performed using KOD-plus-neo DNA polymerase (Toyobo, Osaka, Japan) according to the manufacturer’s instructions. The PCR products were cloned into the pCold-2 expression vector (Takara Bio, Shiga, Japan) at the *Eco*R1 site using an In-Fusion HD cloning kit (Takara Bio, Japan) according to the manufacturer’s instructions. The constructed plasmid was transformed into competent *E. coli* JM109 cells and a single colony was picked from a Luria-Bertani (LB) agar plate containing 100 μg/mL ampicillin and cultured at 37 °C in LB medium supplemented with ampicillin (100 μg/mL) to an optical density at 600 nm of 0.5. The culture was supplemented with isopropyl-ß-d-thiogalactoside to a final concentration of 0.1 mM and cultured at 15 °C for an additional 24 h. The culture was centrifuged at 8000× *g* for 10 min, and the pellet was re-suspended in Tris-buffered saline with Tween 20 (TBS-T: 50 mM Tris-HCl, 150 mM NaCl, 0.1% Tween20, pH 8). Bacterial cells were disrupted by sonication, and inclusion bodies in the cell lysate were washed with TBS-T three times by centrifugation (8000× *g* for 20 min). The inclusion body pellet was dissolved with lysis buffer (6-M guanidine, 50 mM NaH_2_PO_4_, 200 mM NaCl, pH 8) and histidine-tagged protein was trapped with Ni-NTA agarose (Qiagen, Germany) and bound protein was eluted with imidazole and then dialyzed against phosphate buffered saline (PBS). (Electrophoretic image of purified His-tagged recombinant protein was shown in Appendix A). A mouse was immunized with recombinant protein combined with Freund’s complete adjuvant for the first time, and incomplete adjutant for the next three times at 2-week interval. Blood was collected and incubated at 4 °C overnight and centrifuged to collect serum.

### 2.6. Immunohistochemistry

Paraffin sections were prepared from four species of abalone that were experimentally infected and sampled periodically, as described above. For comparison with experimentally infected abalone, sections were prepared of animals fixed with Davidson’s fixative from a group of 8-month-old *H. discus discus* that spontaneously developed abalone amyotrophia in a seed production facility (detailed description in Table 1 footnotes). To confirm the specificity of the antiserum, immunostaining was also performed on sections prepared from paraffin blocks of experimentally infected or healthy *H. discus discus* used for in situ hybridization (ISH) against the MCP gene in a previous publication [18] (Table 1).

Paraffin-embedded tissue blocks were sectioned at a thickness of 5 μm in experimentally infected animals and 3 μm in others, mounted on microscopic glass slides (Matsunami, Japan), and deparaffinized with xylene and absolute ethanol. Sections were blocked with normal goat serum diluted 1:10 in PBS. The slides were incubated with AbALV MCP antiserum or PRV-2 σ1 antiserum diluted 1:1000 in PBS for 60 min at room temperature (20–28 °C). The slides were then washed with PBS, and reaction of the primary antibody was detected using a standard avidin-biotin complex peroxidase (PO) method (VECTASTAIN Universal *Elite* ABC Kit, Vector Laboratories, Burlingame, CA, USA) or using PO-labeled secondary antibody method (Histofine simple stain MAX-PO kit, Nichirei Biosciences, Tokyo, Japan). Finally, the slides were incubated with diaminobenzidine tetrahydrochloride (DAB) substrate chromogen or Histofine simple stain AEC solution (Nichirei Biosciences) for 3 min for color reaction (brown and red, respectively) and counterstained with hematoxylin. To count cells that were positive for immunostaining, sections were observed under a light microscope (B × 51; Olympus, Japan) with a × 40 objective lens and imaged with a CCD camera (VB-7000; Keyence, Japan). The number of positive cells in the 20 images (0.726 mm^2^) of muscle (including shell muscle and foot muscle), gill, and hypobranchial gland, which are the main tissues in which positive cells are observed, was counted. Nerve trunks also often showed fine spot-like positive reactions, and since it was difficult to count the positive cells, only the presence or absence of positive reactions was recorded.

### 2.7. Quantitative PCR

DNA was extracted from approximately 50 mg of tissue using Agencourt DNAdvance (GE Healthcare, Chicago, IL, USA) and eluted with 200 μL of nuclease-free water. The amount of AbALV genome in 1 μL of the eluate was measured by qPCR targeting the MCP gene [18]. Standard curves were constructed using a plasmid vector containing the MCP fragment. The DNA concentration of the eluate was calculated from the absorbance at 260 nm using Nivo (PerkinElmer, Waltham, MA, USA), and the copy number of the MCP gene in 1 μg of total genomic DNA was calculated.

### 2.8. Statistical Analysis

The MCP gene copy number per microgram of total DNA and number of MCP-immunopositive cells were expressed as mean ± standard error. Statistical analyses were performed using software package Statcel3 (OMS, Tokyo, Japan). Multiple comparisons were made with the Steel-Dwass test. Comparisons of virus load between shell muscle and visceral muss were made using the Student’s *t*-test. Cumulative mortality was analyzed statistically with Fisher’s exact tests. A *p* value of <0.05 was considered significant.

## 3. Results

### 3.1. Immunohistochemistry for a Previously Reported Diseased Abalone and Spontaneously Infected Abalone

No positive cells were observed in healthy abalone or by immunostaining with PRV σ1 antiserum (Figure 2). In a paraffin block specimen of diseased abalone subjected to ISH in the previous publication [18], positive cells were dispersed in the muscle (Figure 3A,B) and connective tissue around the intestine (Figure 3C). No positive cells were observed in healthy abalone or by immunostaining with PRV σ1 antiserum (data not shown).

The histopathological characteristics of the disease and the distribution of MCP-immunopositive cells were observed in spontaneously infected *H. discus discus* from seed production (see detailed description in Table 1 footnotes). Visual observation of spontaneously infected abalone showed no shell incisions, which is one of the symptoms of the disease; however, brown pigmentation was observed in the nacreous layer, which is also a symptom of the disease [10]. Histopathological abnormalities were observed in the gills of all 11 abalones. The gills of the diseased abalone were thickened by hyperplasia, the epithelial layer had cavities that appeared to be edema, and the epithelial arrangement was disorganized, resulting in an uneven surface (Figure 3D). In all observed abalone, abnormal cell masses, one of the characteristics of the disease [10], were observed in the gill vascular spaces and connective tissues (Figure 3E), and necrotic cells showing pyknotic nuclea were occasionally observed. The gills of healthy *H. discus discus* showed no epithelial disorganization or cell accumulation (Figure 3F). Abnormal cell masses were also found in the mantle, gastropod, and epipodium of six abalones (Figure 3G). Immunohistochemical observations showed that positive reactions were mainly observed in gill epithelial cells, with occasional positive cells in the cells of the vascular cavity or connective tissue (Figure 3H). Aside from gills, abnormal cell masses containing positive cells were found in some samples of mantle, gastropod, and epipodium of two animals (Figure 3I), while one abalone did not show a positive reaction despite the appearance of many abnormal cell masses in these tissues. No abnormal cell masses appeared in the nerve tissue, and no positive reactions were observed by immunohistochemistry. No organisms that can be confirmed by optical microscopy or naked eye observation, such as bacteria or parasites, were observed.

### 3.2. Infection Experiment 1: Cohabitation Infection with Four Species at Different Ages

#### 3.2.1. Infection Experiment 1: Mortality and Appearance

Two of the 20 *H. gigantea* used as the source of infection died during the 3 days they were kept in the upstream tank.

In 7-month-old *H. gigantea* and *H. discus discus*, mortality in the infected group began to be noticeable about 1 month after experimental infection, and both cumulative mortality rates were significantly higher than in the negative controls (Figure 4A,B). Two of the 20 *H. discus discus* sampled on day 63 of infection showed notches and brown pigmentation on the shell. In *H. diversicolor* at 7 months of age, there were no significant differences in mortality between the infected group and the negative control group (Figure 4C).

In the 4- to 5-year-old abalones, 2 out of 20 animals died for each of *H. diversicolor* and *H. discus hannai* infected groups and the *H. diversicolor* negative control group. There were no significant differences in mortality between the infected and negative control groups for any species.

#### 3.2.2. Infection Experiment 1: Viral Load

Mean AbALV MCP gene copy number in the muscle of the 18 surviving individuals of *H. gigantea* at the end of the 3-day rearing period in the upstream tank as the source of infection was 4.8 × 10^4^ ± 3.0 × 10^4^ copies/μg total DNA.

In 7-month-old *H. gigantea*, viral load increased after 3 dpi and reached a maximum at 56 dpi (Figure 4D).

For the 7-month-old *H. discus discus* treatment group, the water supply to the sampling tank was accidentally stopped after 42. Therefore, sampling was stopped after 42 days. The viral load gradually increased from 3 dpi, and the copy number of the MCP gene reached a maximum at 42 dpi (Figure 4E).

In 7-month-old *H. diversicolor*, virus was detected only on 3 and 7 dpi (Figure 4F).

For *H. gigantea* and *H. discus discus*, the viral loads in 4- to 5-year-old abalone at 63 dpi were significantly lower than those in the 7-month-olds (Figure 5). The amount of virus in the muscle of *H. discus hannai* was significantly higher than that in the muscle of *H. gigantea* and *H. diversicolor*. In *H. discus hannai*, the mean viral load was about 10 times higher than in the visceral mass. MCP gene was not detected in the muscle and visceral mass of *H. diversicolor*.

#### 3.2.3. Infection Experiment 1: Immunohistochemistry

MCP-immunopositivity was observed only in 7-month-old *H. gigantea* and *H. discus discus* of the infected group, but not in 4- to 5-year-old *H. diversicolor*. Representative histological images of immunostaining in the infection test are shown in Figure 6. Positive cells appeared mainly in the hypobranchial *gland* (Figure 6A), gills (Figure 6B), muscles (Figure 6C), and nerves (Figure 6D,E). In the hypobranchial *gland*, relatively large cells with dark staining were observed. In the gill, most of the positive signal was seen in epithelial cells. In muscle, immuno-positive cells that morphologically appeared to be infiltrating cells were dispersed in the connective tissue. In the nerve, positive reactions were found in the central region of the nerve tissue, and the positive signals were in the form of dots that were smaller than cells or nuclei. Immunohistochemical observations of gills and muscles were generally consistent with the findings of spontaneous diseased abalones.

In *H. gigantea*, positive cells first appeared in the gills on 3 dpi and reached a maximum on day 35 (Figure 4G). Gill thickening and epithelial disorganization were observed starting 5 weeks post-infection and were observed until the end of the experiment. In muscle, positive cells appeared at 42 dpi and increased thereafter. In the hypobranchial *gland*, positive cells were observed from 7 dpi, although the number of positive cells per area was low, with the highest number of individuals having 4 cells. In the nerves, positive cells appeared between 3 and 35 dpi (Table 2). No abnormal cell masses were found (Table 2).

In the *H. discus discus*, positive cells appeared in the hypobranchial *gland* on the first day of infection (Figure 4H). In the gill, positive cells appeared at 7 dpi and reached a maximum at 35 dpi. Gill thickening and epithelial disorganization were observed starting at 4 weeks post-infection until the end of the experiment. No positive cells were found in muscles. Two individuals had positive signal in nerves at 21 dpi (Table 2). Abnormal cell masses were mainly observed in the gills and muscles of some individuals in the early stage of the infection test (Table 2), although the positive cells were not included in the abnormal cell masses.

There were no MCP-positive cells or histopathological abnormalities in *H. diversicolor* or in 4- to 5-year-old abalones.

### 3.3. Infection Experiment 2: Injection Infection with H. discus discus and H. diversicolor at 12 Months of Age

Since *H. diversicolor* was not infected with AbALV in infection experiment 1, we compared the susceptibility of *H. diversicolor* and *H. discus discus* by injection infection using the filtrate of diseased abalone homogenate as the source of infection. The amount of inoculated virus, as measured by qPCR, was 1075 copies/abalone and 7535 copies/abalone in the infection test using *H. discus discus* and *H. diversicolor*, respectively.

Eight out of 20 of *H. discus discus* died during the 9 weeks of observation, but no *H. diversicolor* died. In *H. discus discus*, the amount of virus in the muscle continued to increase after the infection (Figure 7A), while the amount of virus in the muscle of *H. diversicolor* reached a maximum at 1 h after inoculation and continued to decrease with time. No viral genome was detected in *H. diversicolor* on days 14 and 21 after inoculation (Figure 7B).

## 4. Discussion

The results of this study suggest that AbALV is the causative agent of abalone amyotrophia in *H. gigantea* and *H. discus discus*. The cumulative mortality in 7-month-olds *H. gigantea* and *H. discus discus* ranging from 46% to 68%, respectively, at 63 dpi. The time of onset of mortality coincided with maximum viral load in both species, and immunostaining showed a maximum in MCP-antiserum positive cells in the gill at this time. No other pathogens, such as bacteria or parasites, were detected in histopathological examinations. Histological findings, including immunostaining, showed generally similar findings between abalones with spontaneous amyotrophia in which AbALV was detected by PCR and animals with experimentally induced infection. Therefore, the deaths observed in the infection study were attributable to AbALV. For the first time in this study, we analyzed the dynamics of pathogen load along with mortality and histopathological changes in abalone affected by amyotrophy.

The results of immunostaining suggest that the main infected tissue is the gill. In both spontaneously and experimentally infected *H. discus discus* and *H. gigantea*, MCP antigen was predominantly detected in the epithelial cells of gill. The decrease in the number of infected cells in the gills as mortality became more pronounced is possibly due to the death of infected cells. The relationship between mortality and gill infection requires further study. The number of positive cells in the muscle increased after 42 dpi in the 7-month-old *H. gigantea*. The mechanism of the increase of positive cells in the muscle at late stage of infection in 7-month-old *H. gigantea* remains unknown. In *H. discus discus*, histological changes after 42 days were not observed because the water supply was accidentally stopped. Retesting of *H. discus discus* is necessary to compare the histological changes in both species. Since the abnormal cell masses, which had been considered a feature of amyotrophy, were formed mainly in the nerves, the nerves had been predicted to be the target tissue of the pathogenic virus [8,10,16]. In this study, some animals with positive nerves were detected among the 7-month-old *H. gigantea* and *H. discus discus*; however, the number of positive abalones was not great. Additionally, the abnormal cell masses did not always contain positive cells. Several experimentally infected 7-month-old *H. discus discus* formed abnormal cell masses, which did not contain MCP antigen-positive cells. Immunostaining of three spontaneously infected *H. discus discus* revealed abnormal cell masses in all three animals, but one abalone did not contain any MCP antigen-positive cells despite the formation of many abnormal cell masses. In the other two abalones, only some of the abnormal cell masses contained positive cells. It is true that MCP-positive cells are observed in abnormal cell masses in some cases, and the relationship with this disease cannot be denied, but abnormal cell masses are probably a secondary host response.

In contrast to previous studies [10,22], 7-month-old *H. gigantea* were got infected with AbALV in the present study, resulting in mass mortality. Additionally, the *H. gigantea* used as the source of infection in this study has been shown to experience mortality during the seed production process. However, *H. gigantea* had been considered to be non-susceptible to abalone amyotrophia because abnormal cell masses have not previously been observed, and the disease was not reproduced by inoculation with filtrates of diseased *H. discus discus* homogenates, and because mass mortality in aquaculture was reported to occur at a higher water temperature than in other abalone species [10]. In addition, *H. gigantea* was reported to rarely suffer mass mortality during seed production and is often successfully produced [22]. Why were the results so different in previous studies and this study? In the histopathological examination, abnormal cell masses were never observed in infected *H. gigantea* in this study, and symptomatic incisions on the front margin of the shells or brown pigmentation inside the shell were also never observed. *H. gigantea*, which was used as the source of infection, also did not show any histopathological lesions, including abnormal cell masses and shell abnormalities. Due to the lack of pathological features in *H. gigantea*, it had been thought to be insensitive to this disease. However, the 7-month-old *H. gigantea* used in this study showed a level of susceptibility to AbALV similar to *H. discus discus*, and care should be taken to avoid infection when rearing juveniles. Another possible explanation for the difference in pathogenicity to *H. gigantea* from previous reports is that the virus has mutated and acquired pathogenicity to *H. gigantea*. To clarify this possibility, samples of old diseased abalone should be obtained, and the genomes of the infected viruses compared.

In the infection test with *H. diversicolor*, no mortality occurred and AbALV was hardly detected in test animals. In the injection-based infection study, AbALV in muscle decreased over time and decreased below the detection limit within 2 weeks. These results indicate that the susceptibility of *H. diversicolor* to AbALV is lower than that of the other three species. While all abalone tested in this study are in the genus *Haliotis*, the phylogenetic relationships among these species align with disease susceptibility. *H. discus discus* and *H. discus hannai* are southern and northern subspecies, respectively, *H. gigantea* is a close relative of *H. discus*, and *H. diversicolor* is the most distantly related of the four species [23,24]. The most distant species, *H. diversicolor*, is not susceptible to AbALV, and *H. gigantea,* the next most distant species, does not develop the abnormal cell masses and shell abnormalities characteristic of the disease in *H. discus* and *H. madaka* [10], suggesting a relationship between the phylogenetic relatedness and susceptibility to AbALV. This study shows that different species of abalone have different susceptibility to AbALV, and AbALV may be virulent to species not examined in this study. We did not observe infection of juvenile *H. discus hannai* in this study, and we believe that this species is also infected and killed by AbALV based on its close phylogenetic relatedness as a subspecies of *H. discus discus* and previously reported cases where AbALV genes and MCP antigens have been detected at high levels in cases of mass mortality during seedling production of *H. discus hannai*.

Abalone amyotrophia is a disease of juveniles and causes less mortality in grown abalones [13]. In tests using 4- to 5-year-old abalones, there were no significant differences in cumulative mortality between infected and control groups for any of the abalone species, and the viral load was lower in 4- to 5-year-old abalones compared to 7-month-old ones. Grown abalone are apparently less sensitive to AbALV, but adults may function as reservoirs for AbALV because they do not die even when infected. A closely related virus, ASFV, is highly pathogenic to domestic piglets and juvenile pigs [25] but causes high mortality regardless of the age of the pigs [26]. However, ASFV has low pathogenicity to its natural hosts, warthog (*Phacochoerus africanus*) [27], bush pig (*Potamochoerus* spp.) [28], and ticks (*Ornithodoros moubata*) [29]. In its natural state, ASFV is maintained in the sylvatic cycle of these animals without causing clinical signs of disease [29]. Likewise, while AbALV infection has not been studied in animals other than abalone, the virus may be maintained in nature with adult abalone as carriers. Infection of abalone amyotrophia in juveniles was reported to be established due to co-habitation with field collected spawners [19], suggesting that the wild parent abalone is a likely carrier of the virus. Okada et al. [30] reported that rearing juvenile abalones separately from apparently healthy large abalone can prevent the development of abalone amyotrophia in juveniles. Selecting virus-free spawners by nondestructive testing methods [31] may prevent infection from parents to juveniles during seedling production.

## Figures and Tables

**Figure 1 viruses-13-02315-f001:**
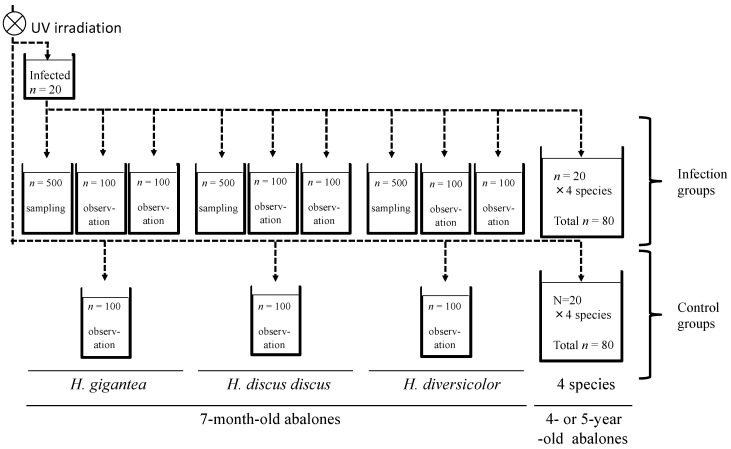
Experimental set-up for infection experiment 1. Seven-month-old *H. gigantea, H. discus discus*, and *H. diversicolor*, and four- to five-year-old *H. gigantea, H. discus discus, H. discus hannai*, and *H. diversicolor* were reared in UV-irradiated seawater (control) and effluent from infected *H. gigantea* (infected group). Dotted arrows indicate water flow.

**Figure 2 viruses-13-02315-f002:**
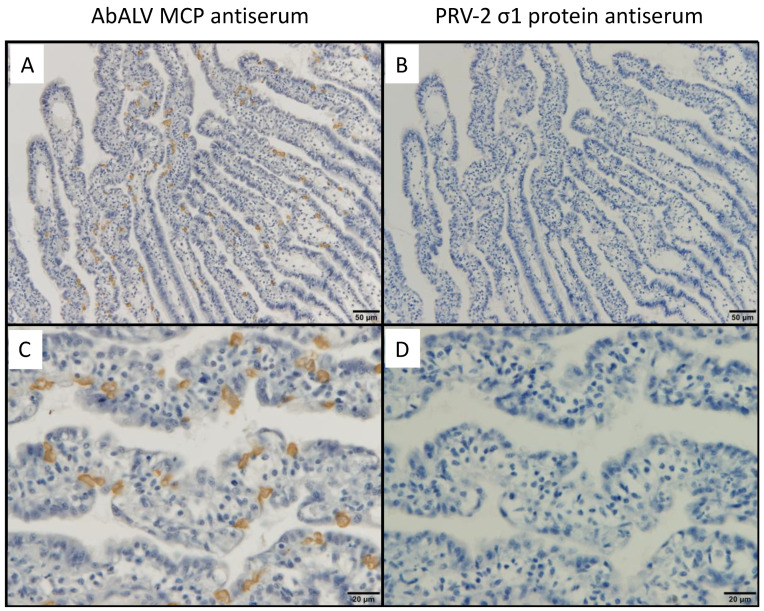
Tissue sections of *H. discus discus* infected with AbALV were immunostained with antiserum against AbALV MCP (**A**,**C**) or against PRV-2 σ1 protein (**B**,**D**) as primary antibody, respectively. Positive cells were found only in staining of diseased abalone with AbALV MCP antiserum.

**Figure 3 viruses-13-02315-f003:**
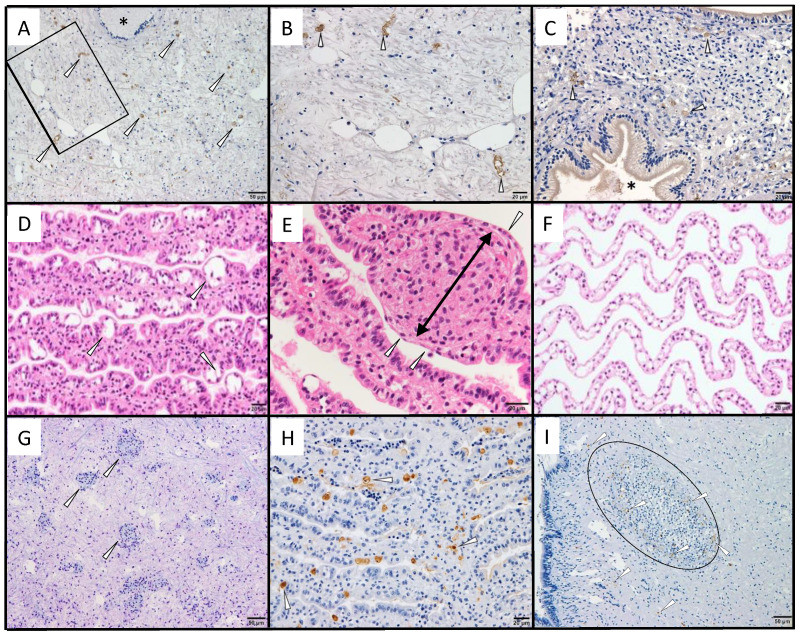
Histopathological images of spontaneously infected *H. discus discus*. (**A**–**C**): Sections were prepared from paraffin blocks in which the MCP gene of AbALV was detected by ISH in a previous publication [18], and immunostained with antiserum against MCP. (**A**): Positive cells in the connective tissue of muscle (arrowhead). Nerves (*) were negative for immunostaining. (**B**): Magnified image of the frame shown in A. Arrowheads indicate positive cells. (**C**): Connective tissue around the gastrointestinal tract. Arrowheads indicate positive cells, and an asterisk indicates the gastrointestinal tract. (**D**): Gill lesion, showing a high percentage of hydrocephalic voids in the epithelial cell layer of the gill (arrowheads) and uneven surface of the gill. (**E**): Cell masses appearing in the gill, showing accumulation of cells in the connective tissue of the gill (black arrow). The cell height of the epithelial cells is lowered (arrowheads). (**F**): Gills of a healthy animal. Cell height of epithelial cells is lower than that of diseased abalone, and epithelial cells are evenly arranged. (**G**): Abnormal cell masses appearing in the gastropod (arrowheads). (**H**): MCP-immunopositive cells observed sporadically in the gill; mainly epithelial cells were positive (arrowheads). MCP-immunopositive cells were also visible in the connective tissue and vascular cavity (arrowheads). (**I**): MCP-immunopositive cells (arrows) in the abnormal cell mass (oval) of the foot muscle. The left side of the picture is out of body.

**Figure 4 viruses-13-02315-f004:**
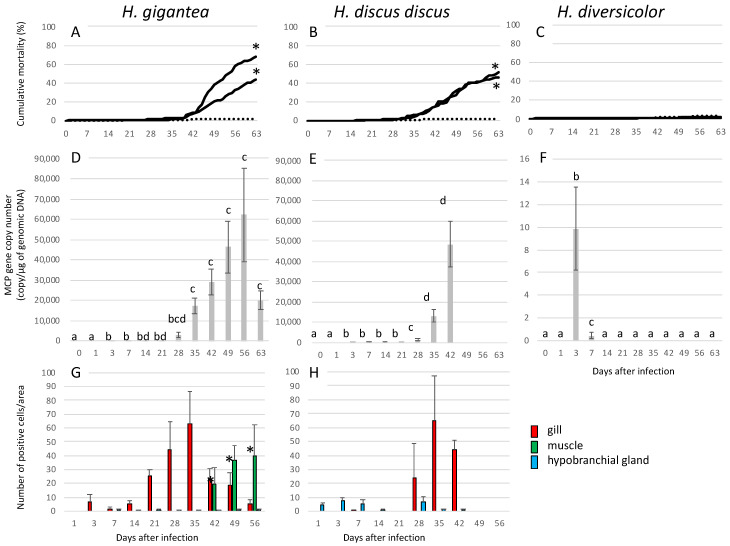
Kinetics of cumulative mortality, numbers of the AbALV MCP gene and immunopositive cells in experimentally infected 7-month-old *H. gigantea, H. discus discus* and *H. diversicolor*. (**A**–**C**): Bold and dashed lines indicate cumulative mortality in the challenged and control groups, respectively. The two bold lines show the duplicated test results. Asterisks indicate significant difference compared to control group (Fisher’s exact test). (**D**–**F**): Copy number of the AbALV MCP gene in the full body, excluding the midgut. Results are expressed as copy numbers per microgram of total DNA. (**G**,**H**): Number of the MCP-immunopositive cells. (**A**,**D**,**G**): *H. gigantea*, (**B**,**E**,**H**): *H. discus discus*, (**C**,**F**): *H. diversicolor.* Different letters indicate significant differences (*p* < 0.05, Steel-Dwass test). In a test of significance within tissues, significant differences were found only in muscles at 42 dpi compared to earlier (asterisks) (*p* < 0.05, Steel-Dwass test).

**Figure 5 viruses-13-02315-f005:**
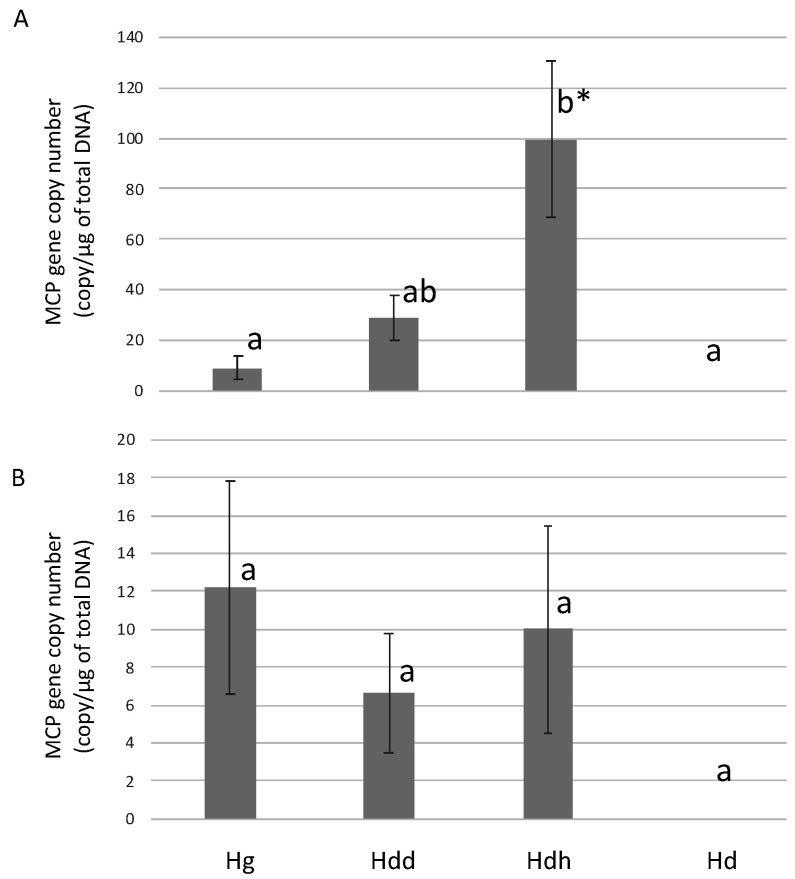
AbALV MCP gene number in the (**A**) foot muscle and (**B**) visceral mass in experimentally infected 4- to 5-year-old *H. gigantea* (Hg), *H. discus discus* (Hdd), *H. discus hannai* (Hddh) and *H. diversicolor* (Hd). Results are expressed as copy number per microgram of total DNA. Different letters indicate significant differences among species (*p* < 0.05, Steel-Dwass test). An asterisk indicates significant differences between organs (*p* < 0.05, Student’s *t*-test).

**Figure 6 viruses-13-02315-f006:**
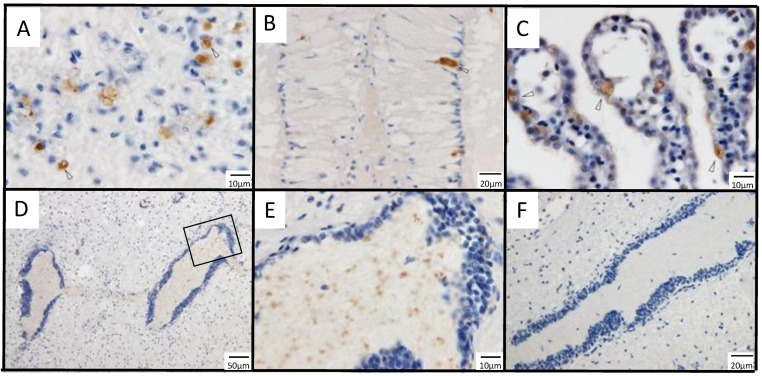
Immunostaining images of 7-month-old *H. gigantea* and *H. discus discus* experimentally infected with AbALV. (**A**): *H. gigantea,* (**B**–**F**): *H. discus discus.* (**A**): muscle, 56 days post-infection (dpi), (**B**): hypobranchial *gland*, 1 dpi, (**C**): gill, 28 dpi, (**D**–**F**): nerve, (**D**): nerve, 21 dpi, (**E**): enlarged image of (**D**), (**F**): A nerve negative for MCP (56 dpi). Immunostaining for comparison with (**D**,**E**).

**Figure 7 viruses-13-02315-f007:**
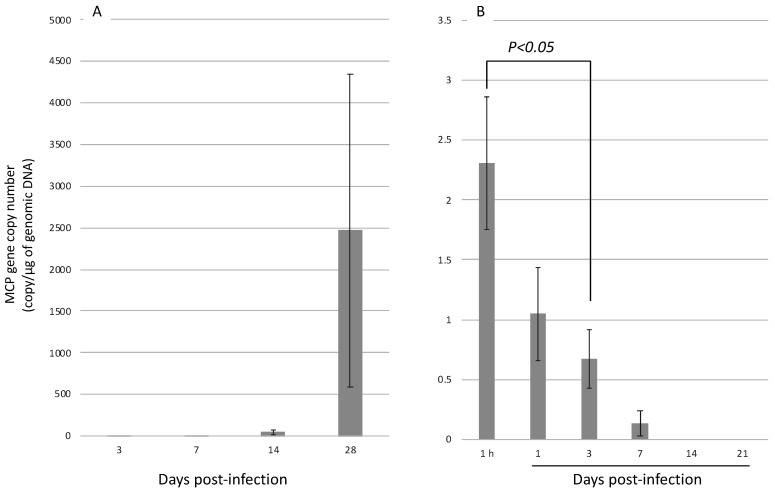
Numbers of AbALV MCP gene in injection challenged 12-month-old (**A**) *H. discus discus* and (**B**) *H. diversicolor.* Results are expressed as copy number per microgram of total DNA.

**Table 1 viruses-13-02315-t001:** Abalone used in this study.

					Body Length (mm)		Body Weight (g)	
Experiments	Species Scientific Name	Starting Disease Status	Use in Experiments	Age at the Start of the Experiment	Range	Mean	Range	Mean
Infection experiment 1								
	*H. gigantea*	Spontaneously infected *	Infectious source	8 months	8–11	9.9 ± 0.3	0.09–0.18	0.15 ± 0.02
		Healthy	Recipient	7 months	7–11	9.7 ± 0.2	0.07–0.17	0.11 ± 0.01
		Healthy	Recipient	4 years	50–61	54.8 ± 1.1	11.0–21.5	15.2 ± 1.0
	*H. discus discus*	Healthy	Recipient	7 months	8–12	10.4 ± 0.3	0.09–0.27	0.16 ± 0.01
		Healthy	Recipient	4 years	46–53	49.0 ± 0.7	10.0–13.9	12.8 ± 0.5
	*H. discus hanai*	Healthy	Recipient	4 years	61–71	64.6 ± 0.8	21.1–32.8	25.6 ± 0.9
	*H. diversicolor*	Healthy	Recipient	7 months	10–17	13.9 ± 0.4	0.15–0.63	0.40 ± 0.03
		Healthy	Recipient	5 years	52–58	52.8 ± 0.8	14.7–30.2	20.1 ± 1.2
Infection experiment 2								
	*H. discus discus*	Spontaneously infected †	Infectious source	13 months	23–25	24.0 ± 0.6	1.3–2.0	1.7 ± 0.2
	*H. discus discus*	Healthy	Recipient	12 months	22–27	23.9 ± 0.5	1.1–2.5	1.5 ± 0.03
	*H. diversicolor*	Healthy	Recipient	12 months	20–27	23.1 ± 0.9	1.0–2.4	1.5 ± 0.03
Immunohistochemistry								
	*H. discus discus*	Experimentally infected ‡		12 months	No record	No record	No record	No record
		Healthy ‡		12 months	No record	No record	No record	No record
		Spontaneously infected §		8 months	16–28	No record	No record	No record
		Healthy		7 months	8–12	10.4 ± 0.3	0.09–0.27	0.16 ± 0.01

*: Cumulative mortality rate in seed production facilities at the time of collection was about 10%. This lot was produced in the same seed production facility at the same time as the spontaneously infected group of 8-month-old *H. discus discus* used for immunohistochemistry (symbol: §). †: Spontaneously infected *H. discus discus*, collected in a previous study [18] and stored at −80 °C until use. ‡: A paraffin block used for in situ hybridization in a previous paper [18] was used. §: Three to four debilitated abalone lacking escape behavior were sampled from each of three tank groups, for a total of 11 abalone for histological observation. Cumulative mortality in the aquariums was 29, 41 and 58% from the onset of mortality. Five individuals from each of these three tanks were sampled and tested for the p72 gene of AbALV using DNA extracted from muscle as a template by conventional PCR [18], and all animals were positive (data not shown). The water temperature at the time of sampling was 20.2 °C. Sections of all abalones were HE-stained for histopathology, and immunohistochemical observations were made on three randomly selected animals.

**Table 2 viruses-13-02315-t002:** Anti-AbALV MCP antiserum positivity in the nerve trunk and appearance of abnormal cell masses by species in 7-month-old abalone in the infected group.

	Number of Abalones with Positive Nerve Trunks (Out of 5 Animals)			Number of Abalones with Abnormal Cell Masses (Out of 5 Animals)		
Days after Infection	*H. gigantea*	*H. discus discus*	*H. diversicolor*	*H. gigantea*	*H. discus discus*	*H. diversicolor*
1	0	0	0	0	1	0
3	2	0	0	0	4	0
7	3	0	0	0	4	0
14	1	0	0	0	3	0
21	2	2	0	0	1	0
28	1	0	0	0	0	0
35	1	0	0	0	0	0
42	0	0	0	0	1	0
49	0	-	0	0	-	0
56	0	-	0	0	-	0

Sampling for *H. discus discus* was limited to 42 days post-infection (dpi) because the water supply to the sampling tank was inadvertently stopped after 42 dpi.

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
