# Peer review of "Susceptibility of Four Abalone Species, *Haliotis gigantea*, *Haliotis discus discus*, *Haliotis discus hannai* and *Haliotis diversicolor*, to Abalone asfa-like Virus"

_viruses, 2021, doi:10.3390/v13112315_

Round 1

Reviewer 1 Report

In the manuscript “Susceptibility of four abalone species, Haliotis gigantea, Haliotis discus discus, Haliotis discus hannai and Haliotis diversicolor, to Abalone asfa-like virus”, the authors examine the susceptibility of various life stages of abalone to asfa-like virus (AbALV).  The authors show that juveniles of H. gigantea and H. discus discus were infectable by AbALV.  However, adults of these species showed low mortality to the disease.  In addition, H. diversicolor is not susceptible to AbALV.  These results suggest that AbALV is the causative agent of abalone amyotrophia.

The experiments were generally well controlled, and the paper was well written.  Specific comments about the manuscript:

  1. I don’t see the asterisks in panel Fig 2C (as described in the figure legend)
  2. Figure 2D, I see two arrows white with a black outline and a black line. The legend says there are white arrows and black arrows.  If the black line is the arrow, should have bigger arrowhead.  Confusing to talk about white arrows when they are white with black outline – is that a white or black arrow?
  3. Figure 2H & I, although the arrowheads are the same, in one figure, they are described as arrowheads and in the other arrows – be consistent in the labelling.
  4. Line 242, says dead cells were occasionally observed. How do the authors know they are dead cells?  Was it visual identification or some other method?  Should clarify the statement.
  5. Typo line 247, “cellss” should be “cells”
  6. Line 252-253 – authors state that no other disease organism that could cause this phenotype was observed. But, what actually did the authors look for?  How have they eliminated the chance that other pathogens might be responsible?  The work here was done using IHC, which would be specific to AbALV and wouldn’t react with other pathogens. 
  7. Figure 3A-C. There are two solid lines and one dotted line.  What do each of the two solid lines represent? It is not clear in the figure legend, it they might be biological replicates, replicates, or something else.
  8. Figure 3E, because water circulation was shut off at day 42, I would stop the experiment at that point and not show additional data points as the experimental protocol was altered and not properly controlled, so there is little value or conclusions that can be drawn after this timepoint.
  9. Figure 4B would be more effective if the scale maxed out at 20 copy/m That would be easier to interpret the variation between samples.
  10. Table 2, was this data impacted by the loss of circulation at day 42? If so, it should be made very clear in all experiments where this data exists.  Better yet, all data points after the water circulation was stopped should be removed from the paper (see point 8)
  11. Comments on line 431-435 are too speculative as controlled experiments weren’t done on this point.

Reviewer 2 Report

The manuscript Authors aim at expanding the current knowledge on African swine fever virus (ASFV)-like viruses, which have been associated to amyotrophia and mass mortality in marine abalone gastropods. The Authors investigated the susceptibility of juvenile and adult abalones of genus Haliotis (H. discus discus, H. discus hannai, H. diversicolor, H. gigantea) to the viral disease, infecting them with the abalone ASF-like virus (AbALV) and subsequently monitoring PCR-detectable virus titers, antisera-detectable viral proteins and animal mortality levels. The Authors confirm AbALV as causative agent of abalone amyotrophia in the tested species, except for H. diversicolor.

If part of a unique trial, the performed work appears huge. Although a large experiment is challenging, I ask if the Authors consider five individuals per species sufficient to describe the AbALV titer within groups (10 or more could better cover the asynchronous infection of individual abalones).

What is the degree of novelty of the manuscript data related to those previously reported (experimental infection of H. discus discus, H. discus hannai in ref.s 9, 10 and 18)?

Can seawater from tanks containing diseased abalones be a sure source of infectant AbALV particles at a titer sufficient to cause disease? Was the AbALV presence or titer regularly measured in the tank seawater?

Some groups of experimentally infected abalones were 7-month juveniles whereas other groups were composed by older adults (I refer to H. discus discus, H. discus hanai, H. diversicolor with 8, 12 months and even 4 and 5 years). For instance, healthy but 5 years old H. diversicolor, finally told not susceptible to infection, have been tested as recipients. What is the purpose to tests adult abalones for susceptibility to infection and disease by AbALV if current knowledge (ref. 13, text line 52) indicates amyotrophia as cause of mortality mainly in juveniles? The abalone body weight values are highly different and heterogeneous, spanning from 0.09 to 32.8 grams. The overall heterogeneity between abalone groups and the measures applied to one or another group pose a doubt on the study design (or on the comparison of independent data series obtained in different conditions). Did the Authors ascertained the PCR-positivity to the virus and histochemical detection of virus proteins on all the abalone groups substantially differing in age and weight? Given possible differences in the abalone life history among groups, comparative conclusions seem more difficult to draw.

In my opinion, Abstract and manuscript text should focus on original, coherent, and valuable findings and contradictory statements should be avoided. For instance, Abstract at line 24 says “Adult abalones had low mortality and viral replication in all species” and at line 27 says “and mass mortality was observed in these species [H. discus discus, H. gigantea]”.

I kindly suggest reconsidering the description of the experimental design, Table 1, Figure 1 and related Methods to make them consistent one to the other and more effective (simpler?) in the whole. For instance, it is not simple to understand when infection was obtained by cohabitation or by injection of a supernant obtained by tissue homogenization of diseased abalones.

Perhaps, the Result section should first focus on the most informative (original and robust) results. Then, Result subsections could specify other (original) pieces of information relevant to the understanding and characterization of the AvALV disease and its management.

About the “Jan-May 2019” H. gigantea abalones found positive to AvALV and dying (ref 18) used as a source of new infection experiments. Have the AvALV-positive abalones been used immediately or stored at -80°C?

About the reasons of H. gigantea susceptibility to AvALV (lines 384-401) it would be important to sequence virus isolates recovered from diseased/dying abalones, to verify the possibility of variant, more virulent, genotypes. Dual transcriptome sequencing during viral infection is currently informative and should be considered because it can unveil the different situation of H. gigantea and H. diversicolor.

Specific changes

L49. “While abalone herpesviruses …[12] have been reported to cause acute..., abalone amyotrophia causes delayed mass mortality mainly in juveniles [13] (possible change)

L72 H.gigantea (check mis-typing)

L84 Please report the document as a complete reference

L151 Please specify the acronym PRV-2, if not previously specified

…and so on.

L294-296. Lack of details impair the overall understanding: what the age of the 18 H. gigantea survivors? In a similar way, please check all the manuscript.

L311-313. Better to avoid qualitative statements (higher) and specify values or value range. Please check the whole text.

Reviewer 3 Report

Please see the manuscript.

Author Response

To the Reviewers

We would like to thank the reviewers for their insightful comments, which have led to significant improvements in our paper. We have appended our response to the PDF file in which you filled in your remarks. Please see the attachment.

Round 2

Reviewer 2 Report

The manuscript has improved. Minor points are the following.

Lines 256 and 286. There are two Figures 2 in the manuscript text, no clear if the first one is text figure or a supplementary figure.

Line 334. Sampling at 42 (not 49) dpi?

Check and correct the Reference style (e.g., at lines 560-563)

Author Response

Thank you for your kind review.
We have made the following corrections to your suggestions.

Lines 256 and 286. There are two Figures 2 in the manuscript text, no clear if the first one is text figure or a supplementary figure.

  In the revised manuscript, Supplemental Figure 2 has been changed to Figure 2, and the numbers of the subsequent figures have been shifted one by one.

Line 334. Sampling at 42 (not 49) dpi?

Corrected as you suggested (line 356 in revised manuscript).

Check and correct the Reference style (e.g., at lines 560-563)

All references have been revised.